# Protective Effect and Mechanism of Melatonin on Cisplatin-Induced Ovarian Damage in Mice

**DOI:** 10.3390/jcm11247383

**Published:** 2022-12-12

**Authors:** Fen Xing, Mengyao Wang, Zhiming Ding, Junhui Zhang, Simin Ding, Lingge Shi, Qinge Xie, Muhammad Jamil Ahmad, Zhaolian Wei, Liang Tang, Dan Liang, Yunxia Cao, Yajing Liu

**Affiliations:** 1Reproductive Medicine Center, Department of Obstetrics and Gynecology, The First Affiliated Hospital of Anhui Medical University, No 218 Jixi Road, Hefei 230022, China; 2NHC Key Laboratory of Study on Abnormal Gametes and Reproductive Tract, Anhui Medical University, No 81 Meishan Road, Hefei 230032, China; 3Key Laboratory of Population Health Across Life Cycle, Anhui Medical University, Ministry of Education of the People’s Republic of China, No 81 Meishan Road, Hefei 230032, China; 4Anhui Province Key Laboratory of Reproductive Health and Genetics, No 81 Meishan Road, Hefei 230032, China; 5Biopreservation and Artificial Organs, Anhui Provincial Engineering Research Center, Anhui Medical University, No 81 Meishan Road, Hefei 230032, China; 6Anhui Provincial Institute of Translational Medicine, No 81 Meishan Road, Hefei 230032, China; 7Key Laboratory of Agricultural Animal Genetics, Breeding and Reproduction, Education Ministry of China, College of Animal Science and Technology, Huazhong Agricultural University, Wuhan 430070, China; 8Faculty of Veterinary and Animal Sciences, Muhammad Nawaz Shareef University of Agriculture, Multan 66000, Pakistan; 9Department of Urology, the Second Affiliated Hospital of Anhui Medical University, Hefei 230601, China

**Keywords:** cisplatin, melatonin, ovarian damage, MAPK pathway, mitochondrial function

## Abstract

Chemotherapeutics’ development has enhanced the survival rate of cancer patients; however, adverse effects of chemotherapeutics on ovarian functions cause fertility loss in female cancer patients. Cisplatin (CP), an important chemotherapeutic drug for treating solid tumors, has adversely affected ovarian function. Melatonin (MT) has been shown to have beneficial effects on ovarian function owing to its antioxidative function. In this research, an animal model was established to explore the effect of MT on CP-induced ovarian damage. Immunohistochemical analysis and Western blot were also used to explore its mechanism. This study reported that MT protects mouse ovaries from CP-induced damage. Specifically, MT significantly prevented CP-induced ovarian reserve decline by maintaining AMH and BMP15 levels. We also found that MT ameliorated CP-induced cell cycle disorders by up-regulating CDC2 expression, and inhibited CP-induced ovarian inflammation by decreasing IL-1β and IL-18 levels. Moreover, MT protected the ovary from CP-induced mitochondrial damage, as reflected by restoring mitochondria-related protein expression. Furthermore, CP caused ovarian apoptosis, as indicated by up-regulated BAX expression. MT was also shown to activate the MAPK pathway. Our results showed that MT could ameliorate ovarian damage induced by CP, implying that MT may be a viable alternative to preserve female fertility during CP chemotherapy.

## 1. Introduction

In recent decades, the survival rate of cancer patients has progressively increased due to advances in chemotherapeutics, resulting in a greater emphasis on preserving the quality of life following cancer therapy [1,2,3]. Having functional ovaries and the ability to have children are crucial aspects of the quality of life for young female patients after cancer therapy. Ovarian function is impaired by chemotherapeutics, which could result in premature ovarian failure and infertility [4,5]. Adjuvants are being sought to protect the ovaries of young female cancer patients undergoing emergency chemotherapy to maintain their fertility. CP is a common chemotherapeutic agent widely used to treat a variety of malignant tumors in the colon, head, lung, neck, uterus and bladder [6]. CP can induce cytotoxic effects by binding to DNA to promote the covalent binding of platinum compounds and purine bases, leading to apoptosis and G2 phase cell block [7]. In addition, previous studies have shown that CP can also induce cytotoxicity in tissues by increasing reactive oxygen species (ROS), inhibiting antioxidant enzyme activity, inducing mitochondrial oxidative stress, and promoting apoptosis, while the continued generation of free radicals in cells is the leading cause of disease [8]. Since CP is not specific to cell selection, it also affects normal cells, leading to physiological disorders in different body systems. The ovary is an important organ that affects women’s fertility. The rate of follicular reserve depletion is influenced by various genetic, hormonal, environmental, and therapeutic factors [9]. Several studies have suggested that CP-based chemotherapy can reduce ovarian follicles in women of reproductive age [10]. Even though numerous molecules and pathways may be implicated in this process, methods to avoid CP-induced loss of ovarian quality remain insufficient. Thus, exploring the mechanism of action of CP on the ovaries, and possible protective agents, is a boon and of good relevance to women with toxic effects such as menopause and premature ovarian failure caused by CP chemotherapy.

MT is mostly secreted by the pineal gland of vertebrates, although it can also be secreted by other organs, including the placenta and ovaries [11]. MT has been shown to increase oocyte quality and fertilization rate, promote follicle size, and postpone ovarian ageing. MT can delay ovarian ageing, regulate ovarian biorhythm, promote follicle formation, and improve oocyte quality and fertilization rate [12]. In addition, MT treatment increases fertility and slows ovarian ageing in mice through the melatonin receptor type 1 (MT1)/AMP-activated protein kinase (AMPK) pathway [13]. MT’s antioxidant properties, synergistic anti-cancer efficacy, and fewer side effects have made it an intriguing candidate as a chemical for chemotherapy in recent years [14]. Animal studies have shown that exogenous melatonin pretreatment reduces cyclophosphamide-induced oxidative stress in the testis, liver, lungs, and kidneys of mice [14,15,16]. By preventing ovarian granulosa cell death and keeping AMH expression stable, melatonin protects against cyclophosphamide-induced primordial follicle loss [17]. Premature ovarian insufficiency (POI) is frequent in young female cancer patients who undergo CP-based chemotherapy. Research shows that by blocking the PTEN/AKT/FOXO3a signaling pathway, MT protects the primordial follicular pool against CP chemotherapy-induced damage [18]. The mechanism by which MT protects against ovarian damage following CP treatment remains unknown.

Therefore, this study aimed to examine whether or not MT could serve as an adjuvant to prevent ovarian damage caused by CP. Our data indicated that MT could alleviate CP-induced ovarian damage by protecting the ovary from ovarian reserve decline, cell cycle disorders, inflammation, mitochondrial damage, and apoptosis.

## 2. Materials and Method

### 2.1. Animals and Ethics Statement

This study was in accordance with the ethical guidelines of the Ethics Committee on Animal Use at the Anhui Medical University. Adult female C57BL/6J mice (*n* = 40), aged seven weeks and weighing about 20 g, were housed in a standard environment. Animals were maintained on a standard diet and water and were free to do as they pleased.

### 2.2. Animal Treatment

After one week of acclimation to the environment, seven-week-old female C57BL/6J mice were enrolled in this study and randomly divided into the following groups (*n* = 10 for each group): Control group (Con), CP alone group (CP), melatonin in combination with CP (CP + MT) group, and melatonin alone (MT). For CP treatment, mice received an intraperitoneal injection with 2 mg/kg CP, repeated on alternate days (dissolved in 0.9% normal saline) until day 21. Control mice received an equal volume of 0.9% normal saline intraperitoneally. For MT treatment, mice received 30 mg/kg MT intraperitoneally for 3 days before and 2 days after CP treatment. Mice were sacrificed 72 h after CP injection. The CP and melatonin dosages were in accordance with a previous study with little modification according to mice [19,20]. All animal experiments were approved by the Anhui Medical University Institutional Ethics Committee (ethics approval number: LLSC20170062).

### 2.3. Immunohistochemical Analysis

Ovarian tissues were washed with 0.9% saline solution, fixed in 4% Paraformaldehyde (Biosharp, Beijing, China), and embedded in paraffin blocks to cut into 4-mm-thick tissue layers, and these layers were fixed on slides. Then after dewaxing in xylene, gradient hydration in ethanol, permeabilization, and antigen repair, the target antibody was selected and closed overnight at 4 °C and washed three times with PBS for 5 min each the next morning. The corresponding secondary antibody was selected and incubated at 37 °C for 2 h. Finally, DAB and hematoxylin staining was performed. The images were photographed with an ortho-fluorescence microscope and analyzed with Image J.

### 2.4. Western Blot

Total protein was extracted by conventional methods, and then boiled for 10 min to denature. The protein (3–5 ul per well) was separated with 10–12% SDS-PAGE. After separation by electrophoresis, proteins were transferred to PVDF membrane. Next, membranes were blocked by 5% milk for 2 h and incubated with the corresponding primary antibodies at 4 °C overnight (Table 1). Next, following incubation with secondary antibodies for about 2 h, membranes were washed 3 times with TBST. Finally, an enhanced chemiluminescence detection system (Thermo Fisher Scientific, Waltham, MA, USA) was used to detect the signals for protein expression levels against ACTIN.

### 2.5. Statistical Analysis

Statistical analyses were performed using Graphpad prim version 8 for Windows. The differences among multiple groups were compared by the one-way analysis of variance (ANOVA). Data were expressed as Mean ± SEM. *p* < 0.05 was considered to indicate statistical significance.

## 3. Results

### 3.1. MT Improved the Expression of AMH and BMP15 after CP Chemotherapy

Anti-Mullerian hormone (AMH), secreted by the granulosa cells of developing follicles, is a valid biological marker of ovarian reserve and is useful for monitoring ovarian function after chemotherapy. After CP treatment, we found that the level of AMH in the ovary was reduced compared to the CN group (Figure 1A,B). However, combination therapy with MT significantly prevented the CP-induced decrease in ovarian AMH levels in the MT + CP group (Figure 1A,B). BMP15, an important oocyte-secreted factor with a leading role in controlling ovarian function in female reproduction, was also investigated in our study. The result showed that BMP15 levels in ovaries were significantly decreased after CP treatment compared to controls (Figure 1E,F). Importantly, MT can ameliorate the down-regulation of BMP15 expression induced by CP (Figure 1C,D). These results suggested that CP depletes ovarian reserve and follicular developmental potential through AMH- and BMP15-mediated pathways and that MT treatment rescued ovarian dysfunction.

### 3.2. MT Ameliorated CP-Induced Cell Cycle Disorders and Apoptosis

To explore the potential protective mechanism of MT against CP-induced ovarian side effects, Western blot assays were performed on ovaries to identify changes in the expression of cell cycle-related proteins (Figure 2A). Although CP treatment did not affect the expression of the cell cycle-associated protein CDC25B and CDK6 (Figure 2B,C), CDC2 expression was significantly reduced in the ovaries after CP treatment (Figure 2A,D), suggesting that CP affected the cell cycle, which is consistent with the results of previous studies. Moreover, after the addition of MT, the expression of CDC2 was normalized (Figure 2A,D), indicating that MT can ameliorate CP-induced cell cycle disorders. In addition, the results showed that the CP treatment could significantly increase cellular apoptosis, as evidenced by the upregulation of BAX expression, which could be rescued by MT treatment (Figure 2A,E).

### 3.3. MT inhibited CP-Induced Ovarian Inflammation

To further investigate the mechanism by which CP affected ovarian function, IL-1β and IL-18 were examined to evaluate the occurrence of inflammation in ovaries. As shown in Figure 3A,B, the expression level of IL-1β protein was significantly higher in the CP treated ovaries than in the control group. Further study showed that MT treatment was able to reduce the cisplatin-induced rise in IL-1β expression (Figure 3A,B). Similarly, MT treatment reduced the CP-induced rise in IL-18 expression by immunohistochemistry analysis (Figure 3C,D), suggesting that MT can abolish CP-induced ovarian inflammation.

### 3.4. MT Protected the Ovary from CP-Induced Mitochondrial Damage

Mitochondria are considered energy factories, essential for cell growth and metabolism. The mitochondria is a dynamically changing organelle, constantly dividing and fusing, and therefore mitochondrial dynamics-related proteins are fundamental to maintaining mitochondrial integrity and function. To investigate the effect of CP on mouse ovarian mitochondria and whether MT can protect mouse ovary by mediating mitochondrial function, we examined the mitochondrial dynamics-related protein OPA1, DRP1, FIS1, and nuclear genes encoding related proteins TOM20 and Mitofilin (Figure 4A). Although CP did not result in changes in the expression of TOM20, Mitofilin and OPA1, significantly decreased FIS1 and DRP1 expressions were observed in the CP-treated mice in comparison with a control group (Figure 4B–F). When MT was co-treated with CP, the expression of Fis1 and DRP1 was normalized, indicating that MT has a protective effect on mitochondrial damage in the ovary caused by CP.

### 3.5. MT Inhibited CP-Induced Apoptosis via the MAPK Pathway

MAPK pathway has previously been associated with cell cycle inhibition. To verify whether CP-induced ovarian damage is associated with the MAPK signaling pathway, we analyzed the expression of pathway-related proteins P-ERK, P-JNK, and P-P38. Moreover, the expression of P-ERK, P-JNK, and P-P38 was elevated after CP treatment, implying that the MAPK signaling pathway was activated (Figure 5A–D). Notably, MT treatment was able to down-regulate the expression of these proteins, restoring them to normal (Figure 5A–D). These results suggest that MT can inhibit CP-induced MAPK pathway.

## 4. Discussion

Most chemotherapy drugs cause reproductive toxicity in women, disrupting physiological homeostasis and adversely affecting many organs, especially in younger patients [21]. Despite the fact that chemotherapeutics can be used to treat numerous types of cancer, ovarian toxicity in young female cancer patients must never be overlooked. A chemotherapeutic agent, CP, is widely used to treat a variety of cancers; nevertheless, due to its delayed metabolic biotransformation and excretion, it promotes ovarian damage in young female patients, resulting in fertility loss. MT is an endogenous hormone, produced mostly by pineal cells, and has substantial endogenous effects on ovarian function regulation. MT is considered to have anti-cancer effects and is therefore considered to be an ideal candidate when applying CP drugs [22]. In this study, we screened the protective effect of MT against CP-induced ovarian damage and explored its mechanisms (Figure 5).

Mitochondria are important organelles in most cells, which not only produce energy, but also regulate cell growth and cell cycle and participate in cell differentiation and apoptosis [23]. In mammals, the mitochondria are maternally inherited organelles, and earlier research has connected mitochondrial dysfunction to aberrant oocytes, maturation failure, and preimplantation embryogenesis [24,25]. Mitochondrial dynamics-related proteins are involved in mitochondrial fusion and division, and are naturally crucial for the ovaries’ development, such as DRP1, OPA1, and Fis1. In our experiments, we found that the expression of DRP1 and Fis1 expression was reduced in mice ovaries after CP treatment, which was consistent with previous studies that CP exerts its adverse effects on the human ovarian cancer SKOV3 cells through mitochondrial dynamics-related proteins [26]. Moreover, DRP1-mediated mitochondrial dynamics were involved in the effect of CP on acute kidney injury [27]. Our study also revealed that the new kinetic-related protein Fis1 was involved in the action of CP on the ovary. More importantly, we observed that MT treatment restored the expression of DRP1 and Fis1, which indicated that MT could protect against CP-induced ovarian damage via alleviating mitochondrial dysfunction. An earlier study has reported that MT attenuates endocrine disruptors or chemotherapeutic drug-mediated mitochondrial dysfunction [28], favoring that melatonin can ameliorate cisplatin-induced mitochondrial dysfunction.

Ovarian damage caused by CP is facilitated by apoptosis and mitochondrial dysfunction, two closely related processes. BAX is one of the most important pro-apoptotic genes, belonging to the Bcl2 family [29]. Therefore, BAX dysfunction can lead to various pathological diseases, including tumors and immune disorders caused by insufficient apoptosis and cardiovascular diseases, neurodegenerative diseases, and viral infections caused by excessive apoptosis [30,31]. Hence, inhibition of excessive apoptosis may reduce ovarian damage. This study provided additional evidence of melatonin’s significant anti-apoptotic impact by demonstrating that it decreased BAX expression in ovarian tissues of the CP + M T group compared to the CP group. Not coincidentally, it has been shown that blocking excessive apoptosis reduces ovarian damage, while its activation in cyclophosphamide-treated rats leads to ovarian dysregulation [32]. Furthermore, we demonstrated the activation of MAPK signaling by MT, which explains the anti-apoptotic effect of MT in terms of molecular mechanism. In any case, our results also proved that melatonin has a powerful anti-inflammatory effect. Compared with the CP group, the CP + MT group significantly reduced the level of inflammatory factors in the ovaries, such as IL-1β and IL-18. Dysregulation of inflammatory factors can lead to various pathological diseases, such as inflammatory diseases and tumors [33]. Hence, inhibition of inflammatory factors may reduce CP-induced ovarian damage. It has been shown that blocking the inflammatory response can inhibit malignant cells in human ovarian cancer [34]. Moreover, other experiments have also demonstrated the anti-inflammatory effects of melatonin [35], and our results further strengthen this view.

Follicle growth requires AMH, produced by the granulosa cells of developing follicles and utilized in the clinic as a diagnostic and predictive marker of female fertility [36]. AMH negatively regulates primordial follicle recruitment by inhibiting primordial follicle activation and reducing the sensitivity of the antral follicles to follicle-stimulating hormone (FSH) during the recruitment phase of follicular dynamics. Thus, AMH is critical to ensure a certain number of primordial and growing follicles in the ovary [37]. Our study found that CP treatment decreases AMH’s level and reduces AMH’s inhibitory effect on primordial follicle recruitment, which may lead to overactivation of the primordial follicular reserve and premature ovarian failure. Fortunately, MT treatment up-regulated the low AMH expression caused by CP, favoring an increase in ovarian reserve function. BMP15 belongs to the TGF-β superfamily; encoding oocytes secreted proteins into the ovarian follicles, where they contribute to creating an environment supporting follicle selection and growth [38]. It has been considered an indicator of developmental follicle potential. In our study, the expression of BMP15 was reduced after CP treatment, indicating impaired follicle development potential. Similarly, MT treatment can rescue CP-induced decrease in BMP15 expression, which suggests that MT protects against cisplatin-induced impairment of follicular developmental potential. In line with our research, a systematic review showed the effectiveness of MT adjuvant treatment in CP in preventing the depletion of ovarian follicles in mice [10]. Furthermore, by reducing ovarian granulosa cell death and sustaining AMH expression, MT protects against cyclophosphamide-induced primordial follicle loss [17]. In the current study, we found a significant reduction in the number of mature ovaries in the CP group compared to the control group. At the same time, the addition of MT treatment partially restored the number of follicles. These data imply that MT may be an effective protective measure for preventing or treating a chemotherapy-induced decline in ovarian reserve. In addition, CP causes mitochondrial dysfunction, which was the first finding in our study, providing an idea to reduce or treat the sequelae caused by chemotherapy. MT has been shown to protect frozen gamete embryos, leading to higher gamete and embryo survival rates [39]. This would suggest that MT could be used as a potential drug to treat chemotherapy-induced fertility dysfunction, which might allow women with impaired reproductive function due to chemotherapy to have better embryos or even regain fertility.

In conclusion, the toxicity of CP to the ovaries can cause infertility by reducing follicular development, increasing the inflammatory response, and inhibiting mitochondrial function, and MT can improve CP-induced ovarian damage (Figure 6). These findings have key implications for fertility maintenance in young cancer patients who undergo chemotherapy. Future studies should explore further which pathway was used by MT to protect against chemotherapy-induced ovarian damage, and evaluate its dosage for clinical application.

## Figures and Tables

**Figure 1 jcm-11-07383-f001:**
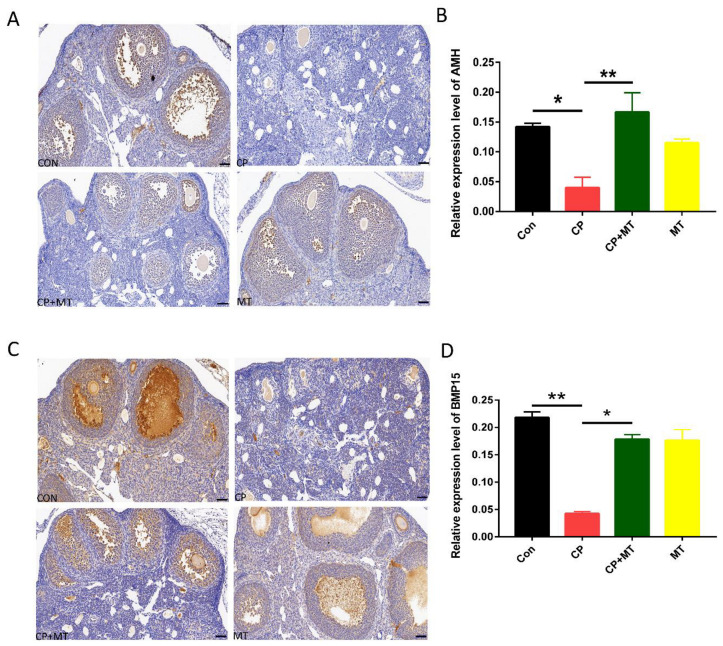
MT improved the expression of AMH and BMP15 after CP chemotherapy. (**A**) Images depicting AMH expression in ovary by immunohistochemistry in Con, CP, CP + MT, and MT groups. Bar = 50 μm. (**B**) The expression level of AMH in ovary in the Con, CP, CP + MT, and MT groups (*n* = 3). *, significant difference (*p* < 0.05). **, significant difference (*p* < 0.01). (**C**) Images depicting BMP15 expression in ovary by immunohistochemistry in Con, CP, CP + MT, and MT groups. Bar = 50 μm. (**D**) The expression level of BMP15 in ovary in Con, CP, CP + MT, and MT groups (*n* = 3). * significant difference (*p* < 0.05). ** significant difference (*p* < 0.01).

**Figure 2 jcm-11-07383-f002:**
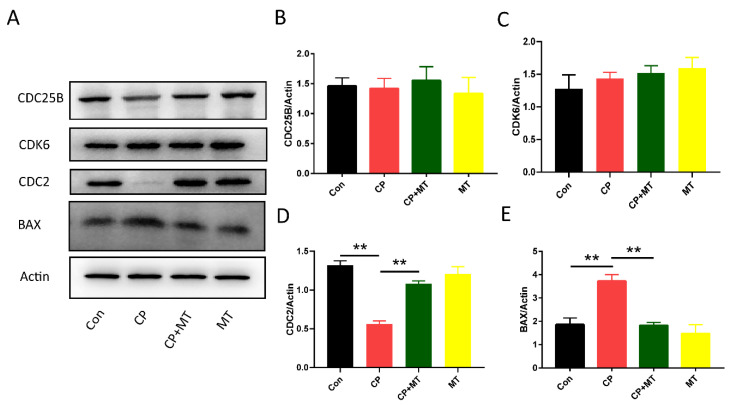
MT ameliorated CP-induced cell cycle disorders and apoptosis. (**A**) Western blot analysis for CDC25B, CDK6, CDC2, and BAX expression in the Con, CP, CP + MT, and MT group (*n* = 3). Actin was used as an internal control to measure the quality of protein. (**B**) Relative intensity of CDC25B was assessed by densitometry. (*p* > 0.05). (**C**) Relative intensity of CDK6 was assessed by densitometry. (*p* > 0.05). (**D**) Relative intensity of CDC2 was assessed by densitometry. ** significant difference (*p* < 0.01). (**E**) Relative intensity of BAX was assessed by densitometry. ** significant difference (*p* < 0.01).

**Figure 3 jcm-11-07383-f003:**
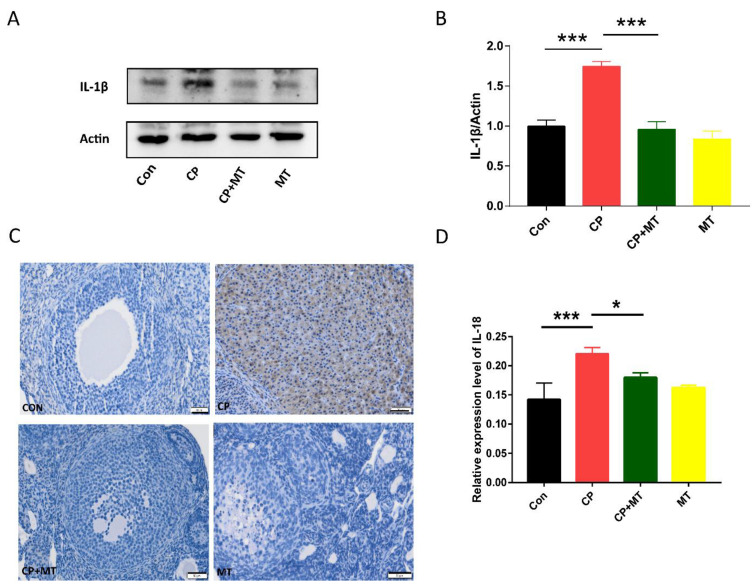
MT inhibited CP-induced ovarian inflammation. (**A**) Western blot analysis for IL-1β expression in the Con, CP, CP + MT, and MT group (*n* = 3). Actin was used as an internal control to measure the quality of protein. (**B**) Relative intensity of IL-1β was assessed by densitometry. *** significant difference (*p* < 0.001). (**C**) Images depicting IL-18 expression in ovaries by Immunohistochemistry in Con, CP, CP + MT, and MT groups. Bar = 200 μm. (**D**) Relative intensity of IL-18 was assessed by densitometry. *** significant difference (*p* < 0.001); * significant difference (*p* < 0.05).

**Figure 4 jcm-11-07383-f004:**
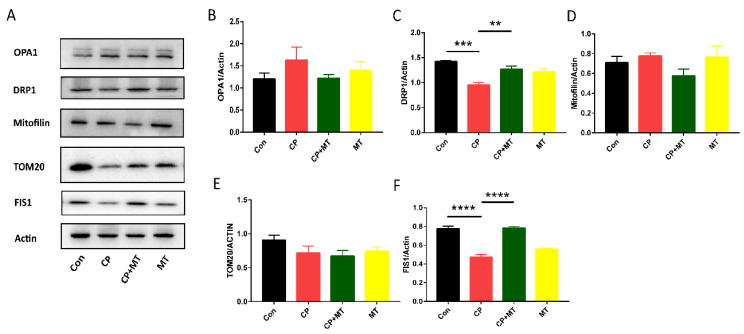
MT protected the ovary from CP-induced mitochondrial damage. (**A**) Western blot analysis for TOM20, Mitofilin, OPA1, FIS1, and DRP1 expression in the Con, CP, CP + MT, and MT groups (*n* = 3). Actin was used as an internal control to measure the quality of protein. (**B**) Relative intensity of OPA1 was assessed by densitometry. (**C**) Relative intensity of DRP1 and actin were assessed by densitometry. ** significant difference (*p* < 0.01).*** significant difference (*p* < 0.001). (**D**) Relative intensity of Mitofilin was assessed by densitometry. (**E**) Relative intensity of TOM20 was assessed by densitometry. (**F**) Relative intensity of FIS1 was assessed by densitometry. **** significant difference (*p* < 0.0001).

**Figure 5 jcm-11-07383-f005:**
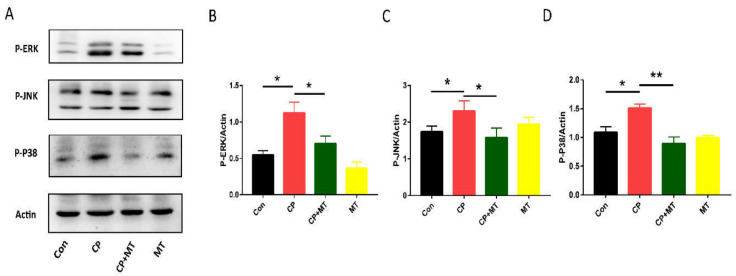
MT inhibited CP-induced MAPK pathway. (**A**) Western blot analysis for P-ERK, P-JNK, and P-P38 expression in the Con, CP, CP + MT, and MT groups (*n* = 3). Actin was used as an internal control to measure the quality of protein. (**B**) Relative intensity of P-ERK was assessed by densitometry. *, significant difference (*p* < 0.05). (**C**) Relative intensity of P-JNK was assessed by densitometry. * significant difference (*p* < 0.05). (**D**) Relative intensity of P-P38 was assessed by densitometry. * significant difference (*p* < 0.05), ** significant difference (*p* < 0.01).

**Figure 6 jcm-11-07383-f006:**
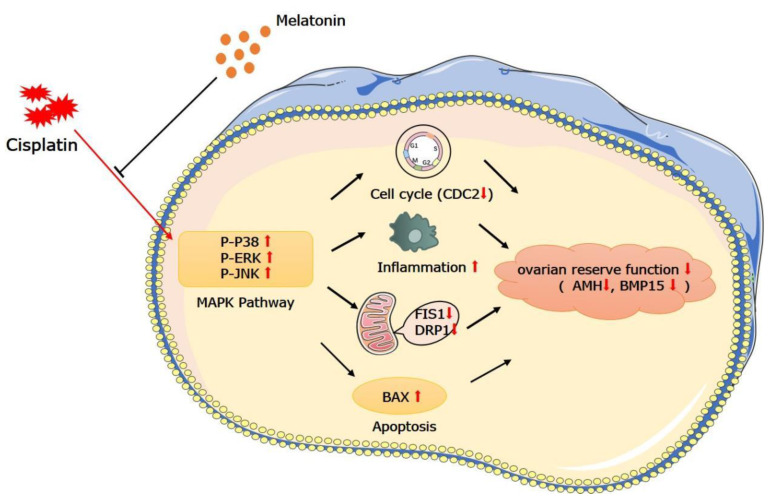
Schematic representation of the protective effect and mechanism of melatonin on CP-induced ovarian damage in mice.

**Table 1 jcm-11-07383-t001:** Detailed information on antibodies.

Antibody	Host spices	Vendor	Catalog No.	Working Dilution
				WB	IHC
Actin	Mouse	ZSGB-BIO	TA-09	1:5000	
Bax	Rabbit	Proteintech	50599-2-Ig	1:2000	
Mitofilin	Rabbit	Proteintech	10179-1-AP	1:2000	
TOM20	Rabbit	Proteintech	11802-1-AP	1:1000	
DRP1	Rabbit	Proteintech	12957–1-AP	1:2000	
Fis1	Rabbit	Proteintech	10956-1-AP	1:1000	
OPA1	Rabbit	Proteintech	27733-1-AP	1:2000	
CDC2	Rabbit	CST	28439S	1:1000	
IL-18	Mouse	Proteintech	10663-1-AP		1:200
IL-1β	Mouse	CST	12426S	1:1000	
AMHBMP15	RabbitRabbit	AbcamProteintech	ab22921218982-1-AP		1:2001:200
P-P38	Rabbit	Proteintech	28796-1-AP	1:1000	
P-ERK	Rabbit	CST	4370S	1:1000	
P-JNK	Mouse	CST	9255S	1:1000

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
