# Peer review of "Protective Effect and Mechanism of Melatonin on Cisplatin-Induced Ovarian Damage in Mice"

_jcm, 2022, doi:10.3390/jcm11247383_

Round 1

Reviewer 1 Report

The study by Xing and collaborators entitled:"Protective effect and mechanism of melatonin on cisplatin-induced ovarian damage in mice" investigated the impact of cisplatin exposure on the structure and function of the ovary in female adult mice and the possible preventive effect of melatonin in the cisplatin -induced ovarian damage. The authors found that melatonin significantly prevented CP-induced ovarian reserve decline by maintaining the level of AMH and BMP, ameliorated cell cycle disorders by up-regulating CDC2 expression and inhibited CP-induced ovarian  inflammation by decreasing IL-1β and IL-18 level. Interestingly, melatonin protected ovary from CP-induced mitochondrial damage, and ovarian apoptosis, as reflected by restoring mitochondria-related protein expression, and up-regulated BAX expression.

I have carefully read the document (especially the results part) and I think the manuscript is interesting. Saying that, however, there are some inconsistencies, and I'd like to strongly recommend the authors to achieve (revise) points listed below.

  1. Material and methods: Section 2.1, “Animals and ethics statement”: Please, provide the approval number given by the ethical committee for this study.
  2. The authors have highlighted the cisplatin and melatonin dose selection according to Kuai et al. [19]. However, this citation just provides the cisplatin dosage. Please specify the melatonin dose used in the current study.
  3. It is not clear whether melatonin was administered previously to cisplatin or after exposure to the latter. I get that the later one is correct but in case of the opposite, i.e. melatonin is administrated first, then it should be a protective role!!!
  4. “Animals and ethics statement”: What were the energy levels and diet composition? How much extra energy was given by melatonin administration? Please explain.
  5. Were animals fasted before the sacrifice? From where was blood obtained? Please add the missing information about sacrifice and collection.
  6. The authors must give more details in the experimental protocol "Immunohistochemical analysis and Western blot.
  7. Western blot: After pooling the proteins of the 3 independent experiments, Western blotting should be performed in (at least) duplicates (which the authors have done), AND a loading control (that could be tubulin, actin or GAPDH) should be given on the same blot for each experiment, and must be used to normalize the signals of target proteins. Nothing of such information was given. The authors should detail this section substantially.
  8. Result: at the two histograms of figure 1 B and D, the author indicates * and **, but at the legend there is only *(P < 0.05). Please check.
  9. Legend of figure 2.C, the author has mentioned **, significant difference (P > 0.05). Please check.
  10. Legend to figures: Please, provide “n” in all figures.
  11. In figure 2.B "CDC25B", the author mentioned no gene variation in the CP group compared to the control. But in western blot there is downregulation of this gene. Please, check.
  12. Check references, reference numbers are repeated
  13. The manuscript has some typo and grammatical errors that need to be fixed.

Author Response

We are thankful to the reviewer for the constructive suggestions on our manuscript. Those comments are really valuable and very helpful for improving our paper. 

Q1. Material and methods: Section 2.1, “Animals and ethics statement”: Please, provide the approval number given by the ethical committee for this study.

A1: Thanks for your suggestion. The approval number (LLSC20170062) has been provided in the revised manuscript.

Q2. The authors have highlighted the cisplatin and melatonin dose selection according to Kuai et al. [19]. However, this citation just provides the cisplatin dosage. Please specify the melatonin dose used in the current study.

A2: It is a very good suggestion. The reference“Yong Zhang et al, Melatonin alleviates acute lung injury through inhibiting the NLRP3 inflammasome, J. Pineal Res. 2016; 60:405–414” mentioned melatonin concentration, and it has been added to the revised manuscript.

Q3. It is not clear whether melatonin was administered previously to cisplatin or after exposure to the latter. I get that the later one is correct but in case of the opposite, i.e. melatonin is administrated first, then it should be a protective role!!!

A3: Thanks for your professional comments. In animal models, melatonin was generally pretreated for 3 days, as in the reference“Yong Zhang et al, Melatonin alleviates acute lung injury through inhibiting the NLRP3 inflammasome, J. Pineal Res. 2016; 60:405–414”.

Q4. “Animals and ethics statement”: What were the energy levels and diet composition? How much extra energy was given by melatonin administration? Please explain.

A4: Thanks for your suggestion. Adult female C57BL/6J mice (n = 40), aged seven weeks and weighing about 20 g, were housed in a standard environment. Animals were maintained on a standard diet, and food and water were provided ad libitum.. We didn't measure how much extra energy was given by – melatonin administration,. This study focused onthe protective effects of melatonin.

Q5. Were animals fasted before the sacrifice? From where was blood obtained? Please add the missing information about sacrifice and collection.

A5: Thanks for your advice. Mice did not fast before death, and no blood was collected because no blood-related tests were involved in the study.

Q6. The authors must give more details in the experimental protocol "Immunohistochemical analysis and Western blot.

A6: Thanks for your suggestion. The Immunohistochemical analysis and Western blot protocols were described in detail in the revised manuscript.

Q7. Western blot: After pooling the proteins of the 3 independent ‎experiments, Western blotting should be performed in (at least) duplicates (which the authors have done), AND a loading control (that could be tubulin, actin, or GAPDH) should be given on the same blot for each experiment and must be used to normalize the signals of target proteins. Nothing of such information was given. The authors should detail this section substantially.

A7: It is a very good suggestion. Actin was used as a loading control. The relative information about actin has been added in the revised manuscript.

Q8. Result: at the two histograms of figure 1 B and D, the author indicates * and **, but at the legend there is only *(P < 0.05). Please check.

A8: Thanks for your suggestion. “**, significant difference (P < 0.01). ” was added in the figure legend in the revised manuscript.

Q9. Legend of figure 2.C, the author has mentioned **, significant difference (P > 0.05). Please check.

A9: Sorry for our carelessness. It has been corrected in the revised manuscript.

Q10. Legend to figures: Please, provide “n” in all figures.

A10: Thanks for your suggestion. It has been added to the revised manuscript.

Q11. In figure 2.B "CDC25B", the author mentioned no gene variation in the CP group compared to the control. But in western blot there is downregulation of this gene. Please, check.

A11: Thank you for your good suggestion. We have checked it carefully, and found there was no significant difference according to  western blot results from three biological replicates

Q12. Check references, reference numbers are repeated

A12: Thanks for your advice. The repeated references have been removed in the revised manuscript.

Q13. The manuscript has some typo and grammatical errors that need to be fixed.

A13: Thanks for the kind reminder! The typo and grammatical errors have been corrected in the revised manuscript.

Reviewer 2 Report

In this study the authors aimed at evaluating the pivotal role of MT in the protection of mouse ovary from CP-induced damage. The manuscript is clear and well written. The study was design accurately and the results are robust and convincing. I have minor comments for the authors.

1. Please provide more details on the protocol used for MT and CP. How did they choose dosage and timings for both MT and CP administration and co-treatment?

2. The authors should further discuss the clinical relevance and implication of the findings, in light of the protocols used for cancer petients and/or fertility preservation

Author Response

Thanks very much for your positive comments!

Q1. Please provide more details on the protocol used for MT and CP. How did they choose dosage and timings for both MT and CP administration and co-treatment?

A1: Thanks for your advice. The principles for choosing dosage and timings of both MT and CP were from the two references “Yong Zhang et al, Melatonin alleviates acute lung injury through inhibiting the NLRP3 inflammasome, J. Pineal Res. 2016; 60:405–414” and “Kuai, C.P., et al., Corydalis saxicola Alkaloids Attenuate Cisplatin-Induced Neuropathic Pain by Reducing Loss of IENF and Blocking TRPV1 Activation. Am J Chin Med, 2020. 48(2): p. 407-428. ”.

Q2. The authors should further discuss the clinical relevance and implication of the findings, in light of the protocols used for cancer petients and/or fertility preservation

A2: Thank you so much for your comments. The discussion section has been revised to make it more clearand implications of clinical findings in light of assisted reproductive techniques (cryopreservation of gametes and embryo) for fertility preservation has been added in the revised manuscript. 

Round 2

Reviewer 1 Report

The authors followed the comments suggested by the reviewers and therefore the MS has been improved accordingly.